



# Probabilistic Tsunami Hazard Analysis of Batukaras Village as a Tourism Village in Indonesia

Wiwin Windupranata[1], M. Wahyu Al Ghifari[2], Candida A. D. S. Nusantara[3], Marsyanisa Shafa[2], Intan Hayatiningsih[2], Iyan E. Mulia[1,4], Alqinthara Nuraghnia[2]

[1]Research Group of Hydrography, Faculty of Earth Sciences and Technology, Institut Teknologi Bandung, Bandung, 40116, Indonesia
[2]Study Program of Geodesy and Geomatics Engineering, Faculty of Earth Sciences and Technology, Institut Teknologi Bandung, Bandung, 40116, Indonesia
[3]Department of Geomatics Engineering, Faculty of Civil Planning and Geo Engineering, Insitut Teknologi Sepuluh
Nopember, Surabaya, 60111, Indonesia
[4]RIKEN Cluster for Pioneering Research, Prediction Science Laboratory, Japan

*Correspondence to*: Wiwin Windupranata (w.windupranata@itb.ac.id)

**Abstract.** Indonesia's location in the middle of tectonic plates makes it vulnerable to earthquakes and tsunamis, especially in
the megathrust zone around Sumatra Island and the southern part of Java Island. Research shows a seismic gap in southern Java, which poses a potential threat of megathrust earthquakes and tsunamis, impacting coastal areas such as Batukaras Village in West Java, a popular tourist destination. To prepare for disasters, Probabilistic Tsunami Hazard Analysis (PTHA), which focuses on seismic factors, was carried out by modeling tsunamis on 3,348 sub-segments of 4 large megathrust segments in the south of Java Island. Stochastic earthquake modeling to simulate the occurrence of a tsunami from an
earthquake with 6.5 Mw to the highest potential magnitude on each segment. This research shows that PTHA in Batukaras Village reveals varying heights of 0.84 m, 1.63 m, 2.97 m, and 5.7 m for each earthquake return period of 250 years, 500 years, 1000 years, and 2500 years. The dominant threat arises from the West Java-Central Java megathrust segment, emphasizing the importance of preparedness, although the annual probability of tsunamis is generally low. Our study will deepen knowledge of tsunami hazards associated with megathrust activities near Batukaras Village for mitigation planning
and decision-making, and it can become a reference for similar coastal tourist areas.

## 1 Background

Indonesia is located in the convergent boundaries between several tectonic plates, making it prone to earthquakes and tsunamis. The meeting between these plates causes the emergence of a zone called the megathrust zone. Megathrust is a type of fault that occurs in the subduction zone of one tectonic plate forced under another plate. High seismic activity in Indonesia
occurs around Sumatra and southern Java due to the megathrust zone (Koswara et al., 2021; Mulia et al., 2019; Supendi et al., 2023; Windupranata et al., 2020).



Research conducted by Supendi et al. (2023) shows a seismic gap to the south of the island of Java that has a potential source of future megathrust earthquakes. This potential earthquake could generate tsunami heights of up to 20 meters on the south coast of West Java, with an average maximum height of 4.5 meters along the south coast of Java. The annual probability of a
tsunami event with a height of more than 3.0 meters that could cause significant loss of life and damage has a probability of 1-10% in the South of Java (Horspool et al., 2014). An earthquake occurred in Pangandaran Regency on 17 July 2006 with a magnitude of 7.6 Mw that generated a powerful tsunami (Fujii & Satake, 2006; Gunawan et al., 2016; Hanifa et al., 2014). This event resulted in more than 300 deaths, 301 serious injuries, 551 minor injuries, and 156 people missing (Mustafida et al., 2022). Other research shows that there is still a high tsunami potential due to the megathrust fault in southern Java, which
has an earthquake return period of 500 years (Harris et al., 2019). This condition is reinforced by the fact that no major earthquakes have occurred in the past few years; only medium-sized earthquakes (Mw<8) have occurred in the past 100 years (Supendi et al., 2023). This means that megathrust earthquakes still have the potential to occur.

Meanwhile, the southern part of Java has high tourism potential to be developed. One of these tourism potentials can be seen from the many tourist villages located on the southern coast of Java, one of which is Batukaras Village, located in
Pangandaran Regency, West Java (Koswara et al., 2021; Nijman, 2021; Windupranata et al., 2020). Batukaras Village is one of the villages in Indonesia that is often visited by local and foreign tourists. Therefore, the preparedness of this tourist village to face a tsunami disaster needs to be reviewed to prevent casualties.

Tsunami hazard analysis in this tourist village can be done by modeling the tsunami using Probabilistic Tsunami Hazard Analysis (PTHA). The PTHA method is an approach that can estimate the tsunami hazard in a certain period of time in each
area that is likely to be exposed to the hazards of the tsunami disaster (Grezio et al., 2017). This method can analyze tsunami hazards originating from seismic (plate tectonic activity) or non-seismic (volcanic activity, submarine landslides, and other events) factors (Grezio et al., 2017; Salmanidou et al., 2019). However, the scenario of potential tsunami hazards from these non-seismic factors is poor, as the timing and mechanism of their occurrence are difficult to estimate (Grezio et al., 2017). In addition, most tsunamis that occur in Indonesia are caused by seismic factors, i.e., vertical displacement of the seabed caused
by shallow earthquakes in subduction zones (Hamzah et al., 2000).

In this study, tsunami modeling based on seismic factors will be conducted using the parameters of megathrust plate activity directly adjacent to Batukaras Village. The results of the PTHA analysis are expected to characterize the tsunami hazard at the study site, thereby facilitating mitigation planning and decision-making. In addition, the results of the tsunami hazard analysis in this village can be used as a reference for other tourist villages, especially for areas with similar characteristics to
Batukaras Village.

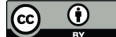


## 2 Data and Method

### 2.1 Data

In this research, accurate bathymetry data will determine the quality of tsunami modeling results. Bathymetry data was obtained from BATNAS (National Bathymetry) provided by the Indonesian Geospatial Information Agency (BIG). This data was used to create the wave propagation domain of the tsunami modeling. Furthermore, earthquake generation parameter data were obtained from the National Centre for Earthquake Studies (PuSGEN) in 2017 and the United States Geological Survey (USGS). The earthquake parameter data is divided into rake, dip, slip, strike, length, and width data obtained from PuSGEN and the depth of the epicenter data obtained from USGS.

### 2.2 Method

PTHA was used as hazard modeling to determine the tsunami risk in Batukaras Village. This method is used to determine the tsunami hazard in an area with a geographically consistent approach to estimating long-term hazards, including uncertainties in the analysis (Grezio et al., 2017) and modeling parameters (Thio et al., 2007). In the process of tsunami analysis using the PTHA method, several stages of data processing must be carried out.

### 2.2.1 Tsunami Green's Function

The calculation of Green's Function aims to obtain the height of the tsunami wave from each observation point. In the calculation of Green's function, domain creation, megathrust sub-segments, and determination of observation points are carried out.

In this tsunami modelling, two types of model domains are formed, namely domain 1 and domain 2. Domain 1 is a large area covering the study area, namely Batukaras Village and the megathrust segment. The megathrust segment used in this tsunami modelling includes the megathrust segments of the Sunda Strait, West Java-Central Java, East Java and Bali (Figure 2). Domain 2 is a smaller area that only covers the study area of Batukaras Village, which is located in Pangandaran Regency.



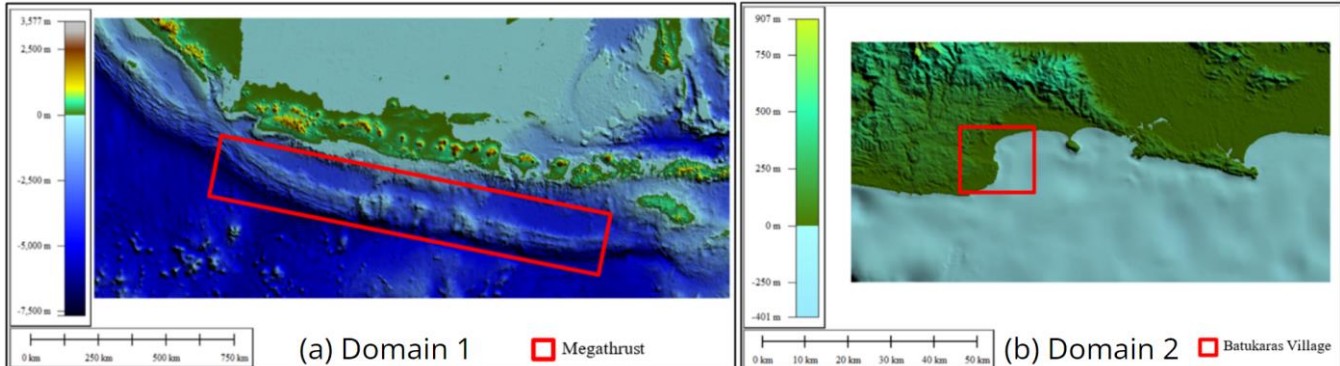

**Figure 1 Modeling domain ilustration (a) Domain 1 (b) Domain 2. Source: BATNAS BIG**

85   In this tsunami modelling, data on the four segments such as location and segment size were sourced from PuSGEN in 2017. Furthermore, the four segments were modelled as megathrust sub-segments with a grid size of 10 kilometres x 10 kilometres, resulting in 3,348 sub-segments divided into:

- Sunda Strait segment (612 sub-segments).

- West Java-Central Java segment (800 sub-segments).

90   - East Java segment (736 sub-segments).

- Bali segment (1200 sub-segments).

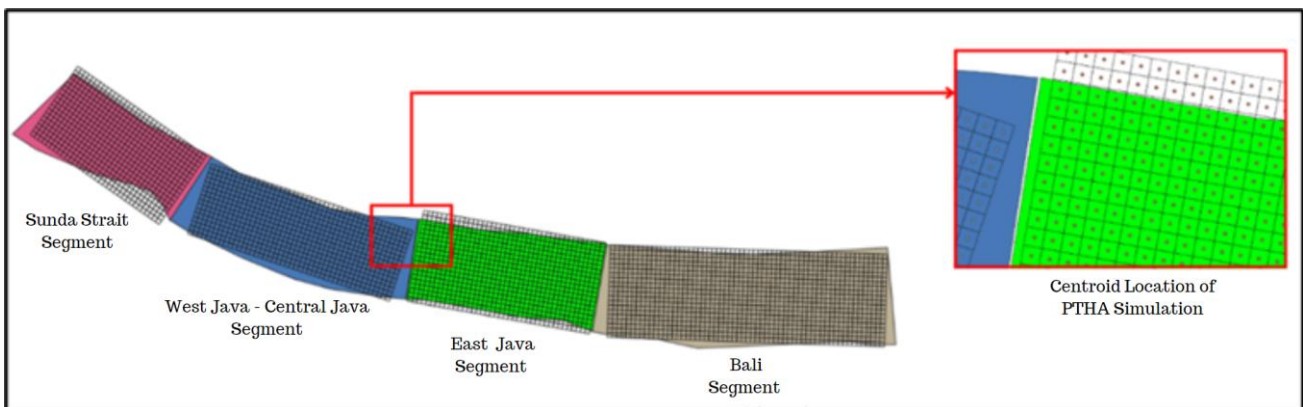

**Figure 2 Megathrust sub-segment used in PTHA Modeling**

In this tsunami modelling, the location of observation points was determined to evaluate the height of the tsunami generated

95   each observation point. The locations of these observation points are spread along the coastline in Batukaras Village. In this modelling, 20 observation points were used based on bathymetry data, where the location of each point has an isobath (depth) of 1 meter.



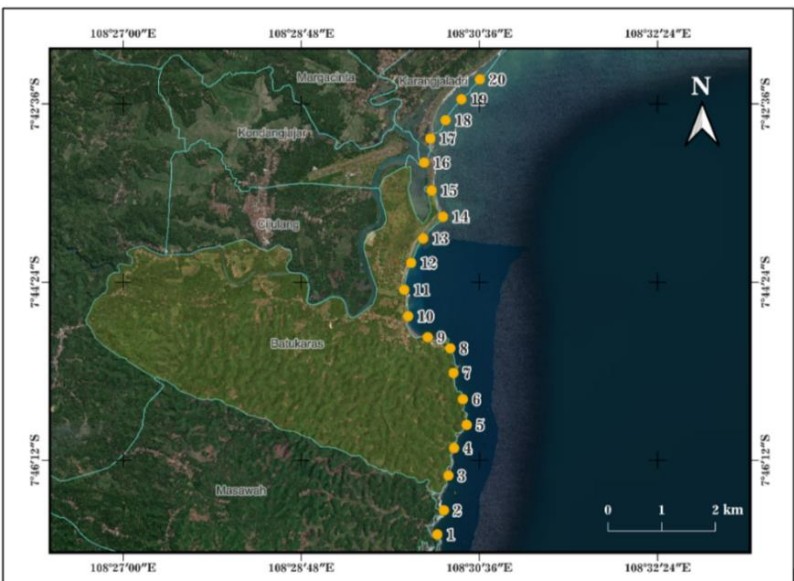

**Figure 3 Observation point along Batukaras coast. Service Layer Credits: Source: ESRI, Maxar, Earthstar Geographics and the GIS User Community**

We used the Cornell Multi-Grid Coupled Tsunami (COMCOT) software version 1.7 for tsunami modelling. In COMCOT, modelling is carried out by generating waves from earthquakes originating from megathrust segments based on earthquake parameter data and propagating the tsunami waves using the shallow water equation to obtain tsunami heights at each observation point. This tool only simulates tsunami waves with no influence of wind and tides.

The COMCOT software uses the Linear Shallow Water Equation (LSWE) and the Non-Linear Shallow Water Equation (NLSWE). The LSWE equation is used when tsunami waves are still in the open sea with smaller wave amplitude compared to depth. The following is the LSWE equation used (Wang, 2009).

$$\frac{\delta\eta}{\delta t} + \frac{1}{R\cos\varphi}\left\{\frac{\delta P}{\delta\psi} + \frac{\delta}{\delta\psi}(\cos\varphi Q)\right\} = \frac{-\delta h}{\delta t} \tag{1}$$

$$\frac{\delta P}{\delta t} + \frac{gh}{R\cos\varphi}\frac{\delta\eta}{\delta\psi} - fQ = 0 \tag{2}$$

$$\frac{\delta Q}{\delta t} + \frac{gh}{R}\frac{\delta\eta}{\delta\psi} + fP = 0 \tag{3}$$

$$\frac{\delta\eta}{\delta t} + \frac{1}{R\cos\varphi}\left\{\frac{\delta P}{\delta\psi} + \frac{\delta}{\delta\varphi}(\cos\varphi Q)\right\} = \frac{-\delta h}{\delta t} \tag{4}$$





$$\frac{\delta P}{\delta t} + \frac{1}{Rcos\varphi}\frac{\delta}{\delta\varphi}\left\{\frac{P^2}{H}\right\} + \frac{1}{R}\frac{\delta}{\delta\varphi}\left\{\frac{PQ}{H}\right\} + \frac{gh}{Rcos\varphi}\frac{\delta\eta}{\delta\varphi} - fQ + F_x = 0 \tag{5}$$

$$\frac{\delta P}{\delta t} + \frac{1}{Rcos\varphi}\frac{\delta}{\delta\varphi}\left\{\frac{P^2}{H}\right\} + \frac{1}{R}\frac{\delta}{\delta\varphi}\left\{\frac{PQ}{H}\right\} + \frac{gh}{Rcos\varphi}\frac{\delta\eta}{\delta\varphi} - fQ + F_x = 0 \tag{6}$$

$$\frac{\delta Q}{\delta t} + \frac{1}{Rcos\varphi}\frac{\delta}{\delta\psi}\left\{\frac{P^2}{H}\right\} + \frac{1}{R}\frac{\delta}{\delta\varphi}\left\{\frac{PQ}{H}\right\} + \frac{gh}{R}\frac{\delta\eta}{\delta\varphi} + fP + F_y = 0 \tag{7}$$

Then, when the tsunami wave travels in shallow waters, the NLSWE equation is used. This is because the wavelength becomes shorter and the wave amplitude becomes larger when passing through shallow water. This means that the shape of the seabed influences the wave amplitude. The following is the NLSWE equation used (Wang, 2009).

$$H = \eta + h \tag{8}$$

$$f = \Omega sin\varphi \tag{9}$$

$$F_x = \frac{gn^2}{H^{7/3}}P\sqrt{P^2 + Q^2} \tag{10}$$

$$F_y = \frac{gn^2}{H^{7/3}}Q\sqrt{P^2 + Q^2} \tag{11}$$

$$P = \int_{-h}^{\eta} u\,dz = u(h + \eta) = uH \tag{12}$$

$$Q = \int_{-h}^{\eta} v\,dz = v(h + \eta) = uH \tag{13}$$

Where g = acceleration of gravity m/s$^2$; P = volume flux in-x (West-East) m/s$^2$; Q = volume flux in-y (South - North) m/s$^2$; f = Coriolis force coefficient; ($\varphi$, $\psi$) = latitude and longitude (°); R = radius of the earth (m); h = water depth (m); $\eta$ = water surface height (m); H = total water depth (m); $\Omega$ = earth rotation rate (7.2921 x 10$^{-5}$ rad/s); (Fx, Fy) = bottom friction at -x and -y; n = manning roughness coefficient (s/m$^{1/3}$); u = current velocity in -x (m/s); v = current velocity in -y (m/s).

## 2.2.2 Stochastic Earthquake Modeling

Stochastic earthquake modelling aims to simulate the slip amount on the fault plane, determining the initial seafloor displacement and the corresponding tsunamis. Mai and Beroza (2002) developed a method to characterise the complexity of earthquake slip represented by spatially random fields of anisotropic wave number spectra according to the von Karman autocorrelation function. The stochastic nature of the method is associated with uniformly distributed random phase angles





embedded in the domain. In this study he random slips are initially calculated at a 1 km grid spacing and then interpolated into the sub-segment size for tsunami Green's function calculation. In this study, bilinear interpolation is performed without changing the average slip, thus maintaining the magnitude of the moment of the interpolated sample. In the formation of the stochastic earthquake model, several settings were used including:

1. The minimum magnitude value used is 6.5 Mw sourced from USGS.

2. The maximum magnitude value used is sourced from PuSGEN in 2017. The maximum magnitude value used is different for each megathrust segment. The Sunda Strait segment has a maximum magnitude of 8.8 Mw; West Java-Central Java of 8.8 Mw; East Java of 8.9 Mw; and the Bali Segment of 9 Mw.

3. The earthquake magnitude bin (interval) used in the modelling is 0.1 Mw, so the earthquake will be generated starting from the minimum magnitude to the maximum magnitude with a difference of 0.1 Mw.

4. The rupture area is randomly specified within the megathrust segment, as done in the PTHA study by Mori et al. (2017).

Figure 4 shows examples of the resulting slip distribution from various earthquake magnitudes by Mulia et al., (2020).

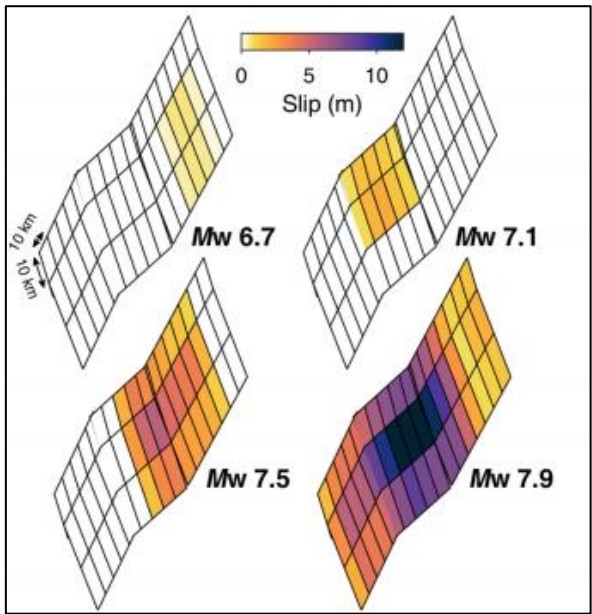

**Figure 4 Stochastic earthquake generation illustration (Mulia et al., 2020)**

We also conducted statistical analyses that show the level of variability in each magnitude range. This analysis uses the
coefficient of variation (CV) = $\sigma/\mu$, where $\sigma$ is the standard deviation and $\mu$ is the mean of maximum coastal tsunami heights at all coastal points. This analysis reflects the convergence of Monte Carlo samples of tsunami heights in coastal areas associated with the sources of the identified active faults, so that the number of samples required across the earthquake magnitude range can be estimated.



We performed this calculation for each increment of magnitude bin (Mw 0.1) of the earthquake in the interval Mw 6.5 to Mw 9.0. The maximum magnitude values used were sourced from PuSGEN in 2017 such as in the Sunda Strait (Mw 8.8), West Java-Central Java (Mw 8.8), East Java (Mw 8.9), and Bali (Mw 9.0) segments. At each magnitude interval, we calculated the coefficient of variation with each increase in the number of samples from 2 to 150.

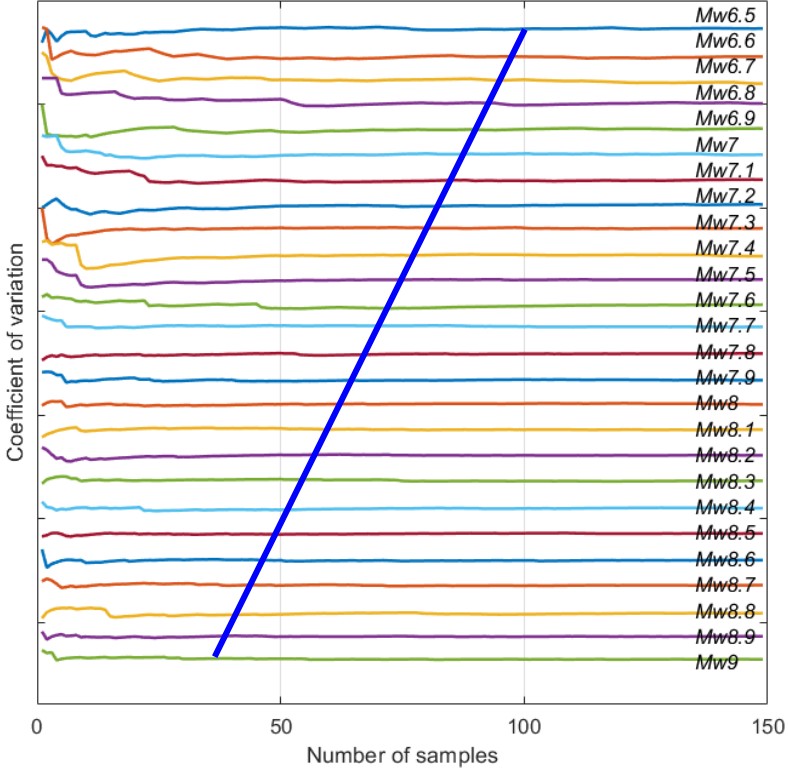

**Figure 5 Coefficient of variation of maximum coastal tsunami height for each magnitude across megathrust segments. The blue line indicates the number of samples required at each magnitude bin for the PTHA**

In figure 5 it can be seen that the variability value decreases as the magnitude increases due to randomisation of the rupture area from smaller earthquakes. Each magnitude shows different variability. For example, the height of a coastal tsunami caused by a Mw 6.5 earthquake is quite stable if about 100 samples are used. Therefore, in this study we assume that the minimum number of samples required is 100 for Mw 6.5. Using a visual approach, we also applied it to all magnitudes by drawing a straight line (blue coloured line). This line shows the number of earthquake scenarios required for each magnitude.

Thus, the resulting number of scenarios is 7,019 scenarios. This large number of scenarios is expected to provide a good explanation of the epistemic uncertainty of the PTHA results in this study.



**Table 1 Total samples at each magnitude bin**

| No. | Mw | Number of faults with magnitude Mw (A) | Number of samples at each magnitude bin (B) | Total samples at each magnitude bin = A×B |
|---|---|---|---|---|
| 1. | 6.5 | 4 | 100 | 400 |
| 2. | 6.6 | 4 | 98 | 392 |
| 3. | 6.7 | 4 | 95 | 380 |
| 4. | 6.8 | 4 | 92 | 368 |
| 5. | 6.9 | 4 | 90 | 360 |
| 6. | 7.0 | 4 | 88 | 352 |
| 7. | 7.1 | 4 | 85 | 340 |
| 8. | 7.2 | 4 | 83 | 332 |
| 9. | 7.3 | 4 | 80 | 320 |
| 10. | 7.4 | 4 | 77 | 308 |
| 11. | 7.5 | 4 | 75 | 300 |
| 12. | 7.6 | 4 | 73 | 292 |
| 13. | 7.7 | 4 | 71 | 284 |
| 14. | 7.8 | 4 | 68 | 272 |
| 15. | 7.9 | 4 | 65 | 260 |
| 16. | 8.0 | 4 | 63 | 252 |
| 17. | 8.1 | 4 | 61 | 244 |
| 18. | 8.2 | 4 | 58 | 232 |
| 19. | 8.3 | 4 | 56 | 224 |
| 20. | 8.4 | 4 | 53 | 212 |
| 21. | 8.5 | 4 | 51 | 204 |
| 22. | 8.6 | 4 | 49 | 196 |
| 23. | 8.7 | 4 | 47 | 188 |
| 24. | 8.8 | 4 | 45 | 180 |
| 25. | 8.9 | 2 | 43 | 86 |
| 26. | 9.0 | 1 | 41 | 41 |
| | | **Total scenario** | | 7019 |

### 2.2.3 Determination of a and b Values Modelling

The values of a and b in PTHA are constants from the empirical formula derived by B. Gutenberg and C. F. Richter with the equation (Gutenberg & Richter., 2010).





$$logN(M) = a - bM \tag{14}$$

Where N is the earthquake frequency; M is the magnitude; a and b are constants.

This equation shows the relationship between earthquake frequency and magnitude. The values of a and b indicate the seismic activity of the megathrust segment, which is influenced by the degree of rock fragility, and these values depend on the observation period, the area of observation, and the seismicity in the area. The larger the value of a for a segment indicates that the segment has high seismic activity; the larger the value of b indicates that the degree of rock fragility in the segment is higher. In this study, the values of a and b were not calculated. Instead, they were sourced from PusGEN in 2017.

### 2.2.4 Probabilistic Tsunami Hazard Analysis (PTHA) Calculation


In this tsunami modelling, the modelling basis is used, which can be seen in the Table 2.

**Table 2 General parameter for Tsunami modeling**

| Parameter | Domain | |
|---|---|---|
| | 1 | 2 |
| Simulation Time (second) | 18,000 | 18,000 |
| Save Time Interval (second) | 60 | 60 |
| Reference Domain | 0 | 1 |
| Grid Size (meter) | 1800 | 600 |

Based on Table 2, the simulation time used for each tsunami scenario run is 6 hours. In addition, earthquake parameter data
sourced from USGS and PuSGEN in 2017 were also determined. Earthquake parameters such as depth, strike, slip, and dip were obtained from the USGS slab 2.0 model, with the rake angle considered opposite to the direction of plate movement in the interseismic phase (Hayes et al., 2018). Meanwhile, the parameters of the epicentre and megathrust segmentation were obtained from the results of the PusGEN study in 2017.

Probabilistic seismic hazard assessment was introduced by (Cornell C. Allin, 1968), which was then adopted in PTHA to
predict the rate of exceedance of a certain tsunami height (H) relative to the tsunami height level (h), which in discrete form can be formulated in equation 15.

$$\lambda(H \geq h) = \sum_{i=1}^{n_s} vi \sum_{j=i}^{n_m} P\big(H \geq h|m_j\big)P\big(M_i = m_j\big) \tag{15}$$

Where, $n_s$ is the total number of i sources/faults; $n_m$ is the number of magnitudes m considered with j intervals; m is the magnitude bin; $\upsilon$ is the occurrence rate of earthquakes from each common plate; and P is the probability of tsunami height.



In this study, a magnitude range from the smallest magnitude to the largest magnitude with a magnitude bin of 0.1 is
considered. The variable υ indicates the occurrence rate of earthquakes with M equal to or greater than each fault calculated
using the Gutenberg-Richter frequency magnitude distribution (Gutenberg & Richter 1944). The variable υ can be defined
by the equation shown in equation 16.

$$v = 10^{a-bm} min \tag{16}$$

Next, the probability of tsunami height is calculated. The probability that the tsunami height H exceeds any tsunami height
level given the magnitude m can be expressed as:

$$P(H \geq h|m) = 1 - \Phi\left\{\frac{ln(h) - ln(H)}{\beta}\right\} \tag{17}$$

With, $\Phi$ is the cumulative standard-normal distribution function; $ln(H)$ is the median logarithmic tsunami height of all
models with a given source and magnitude, and $\beta$ and the standard deviation of $ln(H)$ which can be estimated from the
logarithmic standard deviation. Based on the 2006 Pengandaran tsunami, the value of $\beta$ is 1.0999 (Fritz et al., 2007).

## 3 Result and Discussion

Based on the PTHA that has been carried out, several products are obtained that can be used as material for analysis to
determine the level of tsunami hazard in Batukaras Village.

### 3.1 Tsunami Hazard Curve

The first product of the PTHA is the tsunami hazard curve. The hazard curve is a curve that describes the relationship
between the tsunami intensity value and the return period of an earthquake at an observation point (Grezio et al., 2017). The
tsunami hazard curve for Batukaras Village can be seen in Figure 6.

Based on the hazard curve, the level of hazard can be seen from the probability of tsunami wave heights occurring in
Batukaras Village at each observation point. In this case, twenty observation points were used, spread along the coastline of
Batukaras Village. In addition, the mean and median values of the tsunami height can also be seen. As can be seen from
Figure 6, at an earthquake return period of 100 years, the tsunami height in Batukaras Village does not reach 1 meter, but as
the earthquake return period increases, the tsunami height will increase to 10 meters at an earthquake period of 10,000 years.



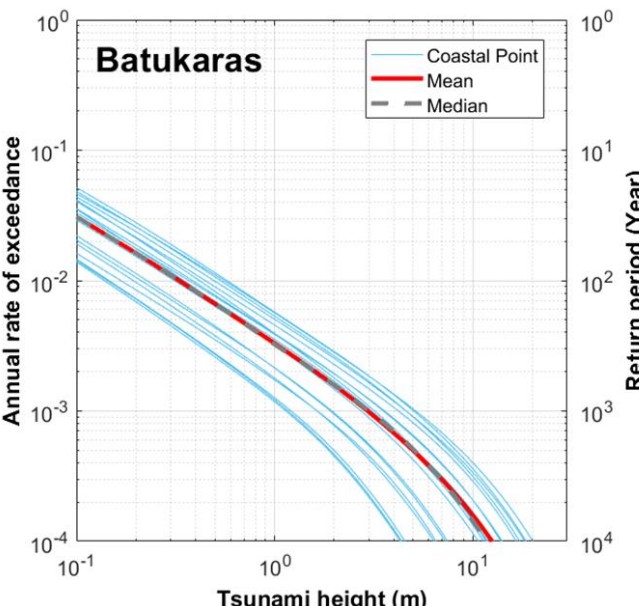


**Figure 6 Tsunami hazard curves at each observation point (light blue) including mean and median value curves in Batukaras Village**

The morphological conditions (shape) of the coast in Batukaras Village can be categorized into two types, namely steep and

sloping coastal areas. When viewed from the distribution of observation points (coastal points) scattered along the coastline of Batukaras Village, the division of the two coastal morphologies can be seen in Figure 7.

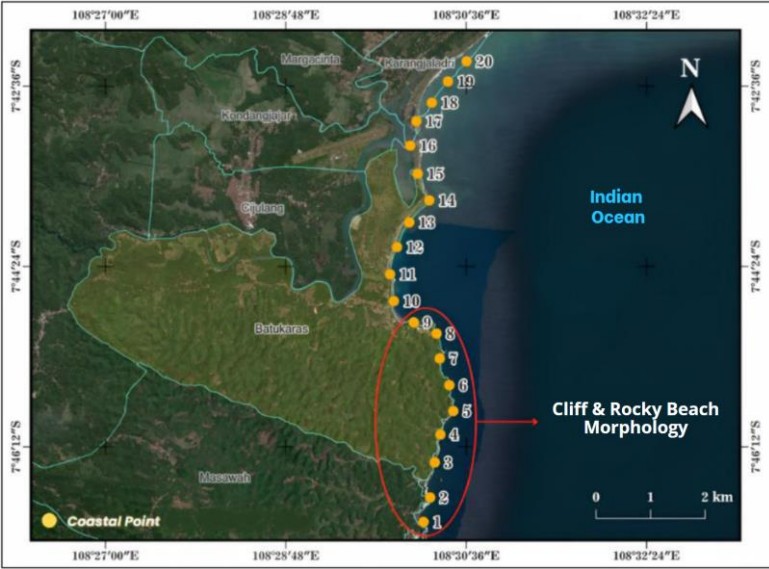

**Figure 7 Beach morphology in Batukaras Village. Service Layer Credits: Source: ESRI, Maxar, Earthstar Geographics and the GIS User Community**





Based on Figure 7, it can be seen that the 1st to 9th observation points are in the coastal area with steep morphology with characteristics of cliff and rocky beach. In contrast, the 10th to 20th observation points are located on the sloping coastal area in Batukaras Village. Therefore, in this PTHA in Batukaras Village, the tsunami height generated from the modeling results of the two types of coastal morphology is identified, which can be seen in Figure 8.

In the resulting hazard curve, a significant difference in average tsunami height can be seen between the two different coastal 215 morphologies in Batukaras Village. In the coastal areas with steep morphology for an earthquake return period of 1000 years, the resulting average tsunami height is still below 10 meters (lower than the average tsunami height in Batukaras Village presented in Figure 6). In contrast, in coastal areas with a gentle sloping morphology, the average tsunami height is already above 10 meters (higher than the average tsunami height in Batukaras Village presented in Figure 6). This shows that the morphology of the coast in Batukaras Village will affect the height of the tsunami generated.


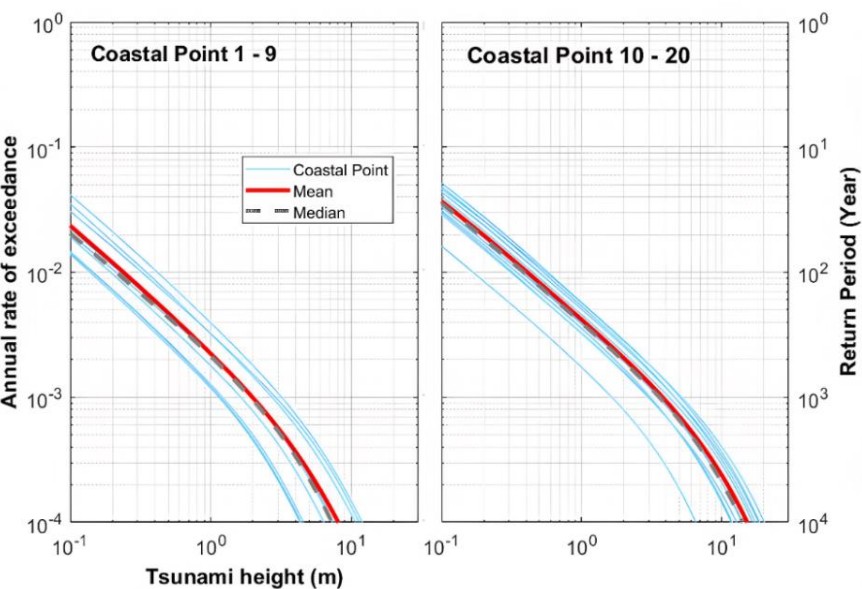

**Figure 8 Tsunami hazard curves based on earthquake return period (a) Observation points 1 to 9; (b) Observation points 10 to 20.**

**3.2 Tsunami Heights Based on Earthquake Return Period**

The second product of the PTHA can also be viewed in graphical form to identify the tsunami height at each observation 225 point. In this case, the tsunami height in Batukaras Village was identified based on four types of earthquake return periods, namely 250 years, 500 years, 1000 years, and 2500 years. This graph can be seen in Figure 9.




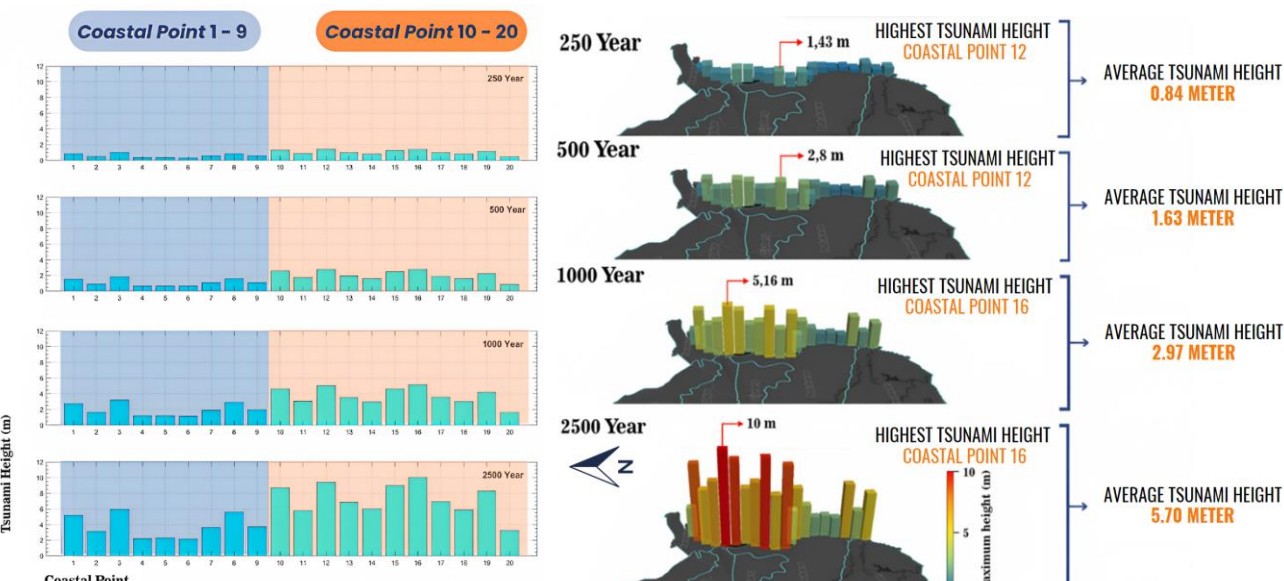

**Figure 9 Visualization of tsunami height based on earthquake return period at each observation point**

Figure 9 shows a graph of the tsunami height at each earthquake return period derived from the tsunami hazard curve in
Figure 6. In this graph, the difference in tsunami height between the steep beach (1st to 9th observation points in the south
area) and sloping beach (10th to 20th observation points in the north area) in Batukaras Village is clearly visible. Overall,
tsunami heights at all observation points located on steep coastal areas were much lower than those on sloping coastal areas
for each earthquake return period.

The average tsunami heights for sloping coastal areas are 0.58 m, 1.1 m, 1.98 m, and 3.75 m for each earthquake return
period of 250 years, 500 years, 1000 years, and 2500 years, respectively. In contrast, the mean tsunami heights for steep
coastal areas are 1.04 m, 2.05 m, 3.77 m, and 7.29 m for each of the 250-year, 500-year, 1000-year, and 2500-year return
periods, respectively. Based on these values, it can be seen that there is a twofold difference in the average tsunami height
between the sloping coastal areas and the steep coastal areas at each earthquake return period.

**3.3 Results of Deaggregation of Hazard in Batukaras Village**

Deaggregation of earthquake hazards will produce the most dominant average magnitude and distance that affect an area
(Aprillianto, 2016). Based on the results of this PTHA, the hazard contribution of each megathrust segment that can have an
influence on tsunami events in Batukaras Village can be known. This deaggregation value can be seen at each observation
point used in this modelling.



Overall, if the hazard deaggregation values are averaged from all observation points scattered along the coastline of Batukaras Village, the tsunami hazard deaggregation in Batukaras Village can be obtained. The hazard deaggregation values of each of these megathrust segments for tsunami events can be seen in Figure 10.

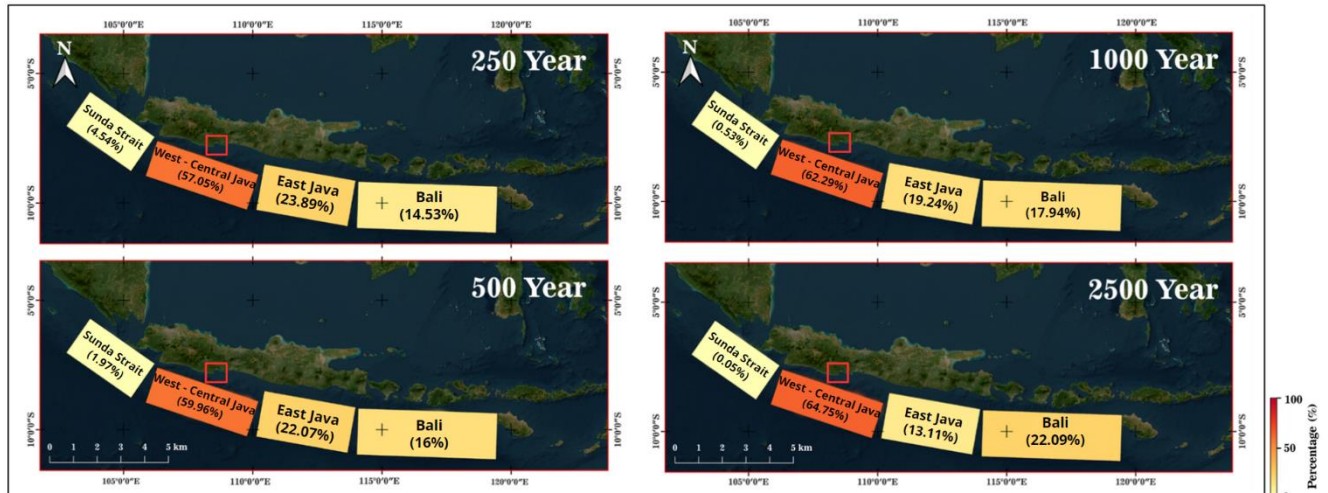

**Figure 10 Deaggregation map of each megathrust segment based on earthquake return period. Service Layer Credits: Source: ESRI, Maxar, Earthstar Geographics and the GIS User Community**

Based on the map, it can be seen the deaggregation of each megathrust segment based on the return period of an earthquake. In each earthquake return period, the deaggregation numbers for each megathrust segment can be seen in Table 3.

**Table 3 Deaggregation of each megathrust segment based on earthquake return period**

| Segments | 250 Year | 500 Year | 1000 Year | 2500 Year | Description |
|:---:|:---:|:---:|:---:|:---:|:---:|
| **Sunda Strait** | 4,54% | 1,97% | 0,53% | 0,05% | **Decrease** |
| **West – Central Java** | 57,05% | 59,96% | 62,29% | 64,75% | **Increase** |
| **East Java** | 23,89% | 22,07% | 19,24% | 13,11% | **Decrease** |
| **Bali** | 14,53% | 16% | 17,94% | 22,09% | **Increase** |

The table shows that the Sunda Strait megathrust segment has the smallest deaggregation for each earthquake return period. This is because the geographical location of this segment is in the western part of Batukaras Village. Thus, tsunami waves generated by an earthquake on this segment would be difficult to travel to Batukaras Village since it does not directly facing the Sunda Straits megathrust. In contrast, the deaggregation of the West Java-Central Java megathrust segment increases with each advancement of the earthquake return period. Geographically, this is because this segment is closest to and directly

faces Batukaras Village. Therefore, if an earthquake occurs with an epicentre originating from this segment, the potential for the tsunami to reach Batukaras Village is quite high.





## 3.4 Results of Annual Probability of Tsunami Occurrence at Earthquake Return Periods

The final PTHA product is the annual probability value of a tsunami occurring in Batukaras Village for each observation point. Based on this, the PTHA results show the annual tsunami probability values for each observation point for tsunami

heights greater than 0.5 m, 1.5 m, and 3 m. These probability values can be seen in Figure 10.

Based on the map, it can be seen that the probability of a tsunami in Batukaras Village is less than 0.1% in any given year. This probability value indicates a very small number of tsunami events in Batukaras Village. Such a result aligns with historical data, in which tsunami disasters are rare in Pangandaran Regency and its surroundings.

The probability for tsunami heights greater than 0.5 m in a given year, with a probability value greater than 0.04%, is only

found at observation points located in sloping coastal areas. The probabilities for tsunami heights of more than 1.5 m and 3 m in a given year are less than 0.02% and 0.001% for all observation points, respectively. However, these probability values should not be used as the main reference because, like earthquakes, tsunamis can occur at any time.

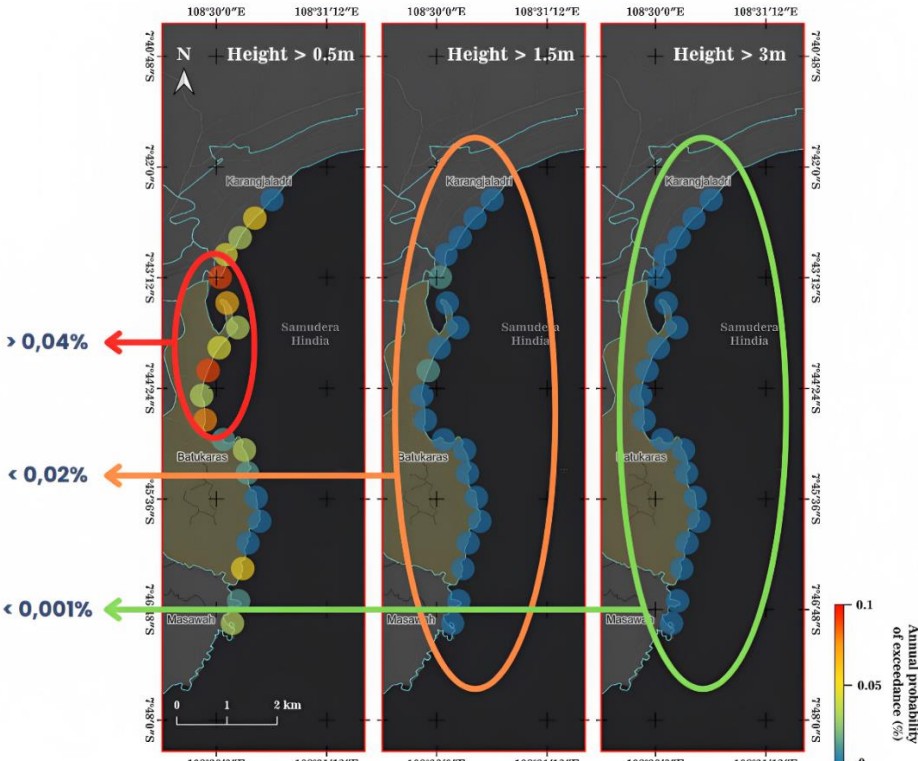

**Figure 11 Annual probability of tsunami occurrence in Batukaras Village based on tsunami height (a) > 0.5m; (b) > 1.5m; (c) > 3m**



## 4 Conclusion

The PTHA in Batukaras Village resulted in tsunami heights of 0.84 m, 1.63 m, 2.97 m, and 5.7 m for each earthquake return period of 250 years, 500 years, 1000 years, and 2500 years, respectively. The results of the tsunami hazard deaggregation in Batukaras Village show that the largest contribution of earthquake sources that can generate tsunamis in Batukaras Village comes from the West Java-Central Java megathrust segment, with a contribution value of more than 57% for each earthquake return period. This can serve as a tsunami warning for Batukaras Village in the event of a high-magnitude earthquake centred on the West Java-Central Java segment. Meanwhile, the annual probability value of tsunami occurrence in Batukaras Village with a height of 0.5 m, 1.5 m, and 3 m has a probability smaller than 0.1% in any given year.

Then, the results of the PTHA can be analysed in more detail by reviewing the tsunami height at each observation point. In the results of the hazard curve, different coastal morphologies produce different tsunami heights. The maximum tsunami height occurs in coastal areas with sloping morphology. This sloping coastal area is very vulnerable because it is dominated by lodging places and restaurants, so it has a high potential to cause casualties in the event of a tsunami. The results of this PTHA modelling can be a reference in making disaster mitigation strategies and scenarios in Batukaras Village, especially for sloping beach areas. This is also considering that Batukaras Village is one of the tourist villages that is visited by many local and foreign tourists.

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
