# Peer review of "Probabilistic Tsunami Hazard Analysis of Batukaras Village as a Tourism Village in Indonesia"

_EGUsphere, 2023_

## Author Response (AR2)

1. The article provides a detailed overview of the PTHA methodology used for tsunami hazard analysis. However, more specific details on the data sources, model assumptions, and validation procedures would enhance the reproducibility and robustness of the analysis. Can the authors elaborate on the process of data validation and verification used to ensure the accuracy of the PTHA results?

   *It is difficult to have validation and verification of the PTHA results, since it is different to a scenario-based tsunami modelling where any evidence based on a tsunami event may available, particularly in the inundation depth. However we try to judge the uncertainties of the results based on PTHA Manual by Thio (2012). A crucial aspect of a probabilistic hazard analysis is the incorporation of uncertainties in both the source and propagation models into the final outcome. There are two types of uncertainties: aleatory and epistemic. These uncertainties belong to frequency and degree of belief approaches to probability respectively. Line 195-211 in the newest version of the paper explain the process.*

2. Although the study acknowledges the uncertainties associated with tsunami hazard modeling, it would be beneficial to include a more in-depth discussion on uncertainty quantification and its implications for decision-making. How were the uncertainties in earthquake parameters, sub-segments size, bathymetry data, and coastal morphology accounted for in the analysis, and how do they impact the reliability of the hazard estimates?

   *We try to judge the uncertainties of the results based on PTHA Manual by Thio (2012). The same as the answer for point 1. Please also refer to Line 195-211 in the newest version of the paper.*

3. The article highlights the vulnerability of sloping coastal areas in Batukaras Village to tsunami hazards. However, a deeper discussion on the specific factors contributing to coastal vulnerability, such as coastline shape, population density, and land use, would provide a better understanding of the potential impacts of tsunamis on the local community.

   *Apart from the sloping coastal factor, the vulnerability of the Batukaras coast to tsunamis is also influenced by the social aspect, where the population in this village is quite dense, reaching 393 inhabitants/km2. Apart from being crowded with residents, the Batukaras coast is also a famous tourist attraction for foreign tourists. This contributes to the vulnerability of the Batukaras coast to tsunami disasters. The coastal area is also vulnerable from an economic perspective, because this coast is dominated by lodging, restaurants, and also agricultural land and plantations, which contribute to the village's regional income. The large amount of land that may be affected by the tsunami could cause significant economic losses for the surrounding community. The results of this PTHA modelling can be a reference in making disaster mitigation strategies and scenarios in*

*Batukaras Village, especially for sloping beach areas. This is also considering that Batukaras Village is one of the tourist villages that is visited by many local and foreign tourists. (Line 307 - 316)*

4.  The results of the PTHA analysis are presented effectively through hazard curves, deaggregation maps, and probability assessments. However, in Figure 8, I suggest to add 95% confidence to present a more comprehensive tsunami height?

    *The request to include 95% of confidence level has been elaborated in Fig. 6 and Fig. 8 of the newest version of our paper.*

5.  The article concludes by emphasizing the importance of disaster mitigation strategies in Batukaras Village. It would be valuable to suggest potential avenues for future research to address knowledge gaps and improve the effectiveness of mitigation efforts. Are there any specific areas of research or data collection that the authors recommend to further enhance understanding of tsunami hazards and vulnerability in Batukaras Village?

    *The results of this study can be used as a reference to conduct further research to calculate economic losses from buildings, land use, and other economic factors from tsunami disasters. In mitigation activities, the results of this research can be used to support the assessment components in preparing Batukaras village to become a tsunami ready village published by IOC-UNESCO considering that Batukaras Village is currently in the recognition process. (Line 321-324)*

**Response on #2 Reviewer's Comments:**

Based on the Probabilistic Tsunami Hazard Analysis (PTHA), this article has presented the complete analysis of tsunami hazard through the numerical modeling of Green's function using COMCOT. This study is focused on the evaluations of seismic factors. The analyzed process of this study is standards and results are reasonable. Only limited print errors and extended discussions are suggested. The revision is considered as minor revision.

1. The study area is a single small village, and for paths from all distant tsunami sources, each analysis point can be considered the same. The main reason for the differences in tsunami hazard curves is the bathymetry near each coastal station and the wave propagation directions of incidence from each tsunami source. This study uses a large amount of computational resources to simulate tsunami propagation, for the reason of tsunami Green's function computed, in the future, it may be possible to simplify the analysis and got the similar result.

   *Thank you for your comments, yes we only study on a small village but the tsunami sources come from different megathrust segments along the southern coast of Java Island. Later on the result can be implemented as well for a large area. Related to analysis we try to have as much as possible interpretation from the simulation results, therefore we used a large amount of computational resources.*

2. Based on the analysis results of this study, it is useful to provide a suggestion for the plan of Tsunami Hazard mitigation in Batukaras Village.

   *It is already included in the new version of the paper in Line 321-324.*
   **The results of this study can be used as a reference to conduct further research to calculate economic losses from buildings, land use, and other economic factors from tsunami disasters. In mitigation activities, the results of this research can be used to support the assessment components in preparing Batukaras village to become a tsunami ready village published by IOC-UNESCO considering that Batukaras Village is currently in the recognition process.**

3. Printed errors:

   Line 41: only medium-sized earthquakes (Mw<8). Mw 8 is a medium-sized earthquake?

   *We have removed the term medium sized, we used the following sentence instead (Line 40-42).*

   **This condition is reinforced by the fact that no major earthquakes have occurred in the past few years; only earthquakes of magnitude below Mw<8 have occurred in the past 100 years (Supendi et al., 2023).**

Line 120: In this study the random slips are initially calculated, he as the.

*It has been corrected already and shifted to line 125 with the following sentence:*
***In this study random slips are initially calculated at a 1 km grid spacing and then interpolated into 125 the sub-segment size for tsunami Green's function calculation.***